# Propolis Has an Anticancer Effect on Early Stage Colorectal Cancer by Affecting Epithelial Differentiation and Gut Immunity in the Tumor Microenvironment

**DOI:** 10.3390/nu15214494

**Published:** 2023-10-24

**Authors:** Ming-Hung Shen, Chih-Yi Liu, Kang-Wei Chang, Ching-Long Lai, Shih-Chang Chang, Chi-Jung Huang

**Affiliations:** 1Department of Surgery, Fu Jen Catholic University Hospital, Fu Jen Catholic University, New Taipei City 243089, Taiwan; a00176@mail.fjuh.fju.edu.tw; 2School of Medicine, College of Medicine, Fu Jen Catholic University, New Taipei City 242062, Taiwan; cyliu@cgh.org.tw; 3Department of Pathology, Sijhih Cathay General Hospital, New Taipei City 221037, Taiwan; 4Taipei Neuroscience Institute, Taipei Medical University, Taipei City 110301, Taiwan; kwchang@tmu.edu.tw; 5Laboratory Animal Center, Taipei Medical University, Taipei City 110301, Taiwan; 6Division of Basic Medical Sciences, Department of Nursing, Chang Gung University of Science and Technology, Taoyuan City 333324, Taiwan; dinolai@mail.cgust.edu.tw; 7Research Center for Chinese Herbal Medicine, Chang Gung University of Science and Technology, Taoyuan City 333324, Taiwan; 8Division of Colorectal Surgery, Department of Surgery, Cathay General Hospital, Taipei City 106438, Taiwan; cgh06719@cgh.org.tw; 9Department of Biochemistry, National Defense Medical Center, Taipei City 114201, Taiwan; 10Department of Medical Research, Cathay General Hospital, Taipei City 106438, Taiwan

**Keywords:** early stage colorectal cancer, propolis, gut immunity, tumor microenvironment, cytokeratin 20, CD4 protein, forkhead box protein P3

## Abstract

Colorectal cancer (CRC) is one of the most common cancers and is the second leading cause of cancer-related death in the world. Due to the westernization of diets, young patients with CRC are often diagnosed at advanced stages with an associated poor prognosis. Improved lifestyle choices are one way to minimize CRC risk. Among diet choices is the inclusion of bee propolis, long recognized as a health supplement with anticancer activities. Understanding the effect of propolis on the gut environment is worth exploring, and especially its associated intratumoral immune changes and its anticancer effect on the occurrence and development of CRC. In this study, early stage CRC was induced with 1,2-dimethylhydrazine (DMH) and dextran sulfate sodium (DSS) for one month in an animal model, without and with propolis administration. The phenotypes of early stage CRC were evaluated by X-ray microcomputed tomography and histologic examination. The gut immunity of the tumor microenvironment was assessed by immunohistochemical staining for tumor-infiltrating lymphocytes (TILs) and further comparative quantification. We found that the characteristics of the CRC mice, including the body weight, tumor loading, and tumor dimensions, were significantly changed due to propolis administration. With further propolis administration, the CRC tissues of DMH/DSS-treated mice showed decreased cytokeratin 20 levels, a marker for intestinal epithelium differentiation. Additionally, the signal intensity and density of CD3^+^ and CD4^+^ TILs were significantly increased and fewer forkhead box protein P3 (FOXP3) lymphocytes were observed in the lamina propria. In conclusion, we found that propolis, a natural supplement, potentially prevented CRC progression by increasing CD3^+^ and CD4^+^ TILs and reducing FOXP3 lymphocytes in the tumor microenvironment of early stage CRC. Our study could suggest a promising role for propolis in complementary medicine as a food supplement to decrease or prevent CRC progression.

## 1. Introduction

Colorectal cancer (CRC) is one of the most common cancers and is the second leading cause of cancer-related death in the world [1,2]. There are significant differences in the incidence and mortality of CRC between different regions, ages, and lifestyles [2,3]. For example, the incidence of people under 50 years of age suffering from CRC is rising globally due to the westernization of diets [4,5,6,7]. The Western diet may have deleterious effects on the gut, such as promoting unfavorable microbes, increasing DNA damage in colon cells, and impeding their DNA repair [4,7]. However, patients, mostly those with early onset CRC, are often diagnosed at advanced stages and thus have poor prognoses [8,9].

Improved lifestyle choices are one way to minimize CRC risk because poor diet increases the risk of its onset [10,11]. However, many naturally occurring products have effective anti-CRC properties [12]. One of these natural products, propolis, is a complex resinous mixture obtained by honeybees, in turn derived from a variety of plant flowers [13,14]. Propolis is recognized as a useful health supplement due to its potential beneficial and nontoxic effects on human health [15,16]. However, propolis also has clinical uses, including for anticancer treatment and as an adjuvant therapy to reduce complications [17,18]. This natural product not only exhibits potent anti-CRC activity by regulating various signaling molecules [19], but also triggers CRC cell death by increasing DNA condensation to reduce the proliferation rate [20,21].

Many studies have highlighted the impact of the tumor immune microenvironment on tumorigenesis, prognosis, and metastasis in CRC [22]. Colorectal tumors often contain prominent immune infiltrates with antitumoral adaptive immunity [23,24]. At all stages of CRC, the immune microenvironment—characterized by different immune responses—might contribute to different outcomes [25,26]. Therefore, gaining an understanding of the effects of propolis on the gut environment, especially intratumoral immune changes, and of its anticancer effect on the occurrence and development of CRC, is worthwhile.

Dysregulated inflammatory responses are a major risk factor for CRC onset [27]. Studies have demonstrated that any regimens or supplements that exhibit strong anticancer, anti-inflammatory, or enhanced immune response can be used to prevent modalities for CRC [28]. It is conceivable that if the inflammation of the intestinal tract were to be suppressed, this would be clinically beneficial. However, there have been no studies on propolis-induced changes to the intratumoral microenvironment of CRC tissues, or on the effect of propolis on CRC cells. In this study, we used X-ray microcomputed tomography (µCT) to visualize colon malformations and tumors in situ in living CRC-bearing mice after propolis administration. We also evaluated the effects of propolis on the intratumoral microenvironment of CRC tissues by using hematoxylin and eosin (H&E) staining and immunohistochemical examination. Here, we have demonstrated that propolis is a useful natural product in alleviating or attenuating primary CRC in an animal model.

## 2. Materials and Methods

Experimental animals and CRC induction by a carcinogen. BALB/c mice aged 7 weeks were purchased from the National Laboratory Animal Center (Taipei, Taiwan) [29,30]. The animal experiment was conducted in compliance with ARRIVE (Animal Research: Reporting of In Vivo Experiments) guidelines on the principles of reduction, refinement, and replacement and was approved (approval no. IACUC 109-024) by the Institutional Animal Care and Use Committees of Cathay General Hospital, Taipei. Mice (3–5 per cage) were housed in an individually ventilated cage rack system (Tecniplast, Varese, Italy) under the following conditions: 50 ± 10% humidity, 12/12 h light/dark cycle, at 23 ± 2 °C. A colonic carcinogen 1,2-dimethylhydrazine (DMH; cat. no. D0741, Tokyo Chemical Industry Co., Tokyo, Japan) and sodium dextran sulfate (DSS, 40 kDa; cat. no. D5144, Tokyo Chemical Industry Co.) were used to induce local colon tumors. Propolis (Promunel Propolis ESIT6; B Natural, Corbetta, Italy) was used as a supplement in the experimental mice. Briefly, mice were quarantined for the first 7 days then randomly allocated as follows: (Group I) control group (*n* = 2), comprising mice that received no treatment; (Group II) DMH/DSS group (*n* = 6), comprising mice that received DMH through intraperitoneal injection and DSS in their drinking water, but no propolis; and (Group III) DMH/DSS/propolis group (*n* = 3), comprising mice that received DMH/DSS and propolis.

The times for injecting DMH (40 mg/kg body weight), drinking deionized water or aqueous DSS (3%), monitoring body weight, eating propolis (30 mg/mouse per time) [31], performing µCT, and final sacrifice are shown schematically in Figure 1A. Mice were euthanized with CO_2_ in a cage when they showed weakness and rapid weight loss of 15–20% at the end of this experiment. The CO_2_ flow rate was set to displace 30% of the cage volume per min. The criteria to confirm death were immobility for more than two minutes and lack of spontaneous breathing. Colons were removed by dissection, rinsed with ice-cold 0.9% NaCl, placed on filter paper, and their lengths measured by ruler. The duration of this experiment was about five weeks.

In vivo µCT and image analysis. Animals were anesthetized with 1% isoflurane, and 30 mg/kg Baritop LV (150 µL, Guojien Co., Taichung, Taiwan) was probably administered before anesthesia if taken orally. Three hours after Baritop administration, the mice were positioned supine in the sample holder of the µCT (Skyscan 1176, Bruker micro-CT, Kontich, Belgium) and immobilized with medical tape to reduce motion artifacts. Scan conditions were as follows: 50 kV source voltage, 0.2 mm aluminum filter, 400 μA source current, exposure time 100 ms, 35 μm isotopic resolution, 1 projection image per 0.7° gantry rotation step, rotation range 360°, with a field of view covering all abdominal regions through to the anus.

Volume data were reconstructed using NRecon software v2.0 (Bruker micro-CT). Image analysis of the mouse abdominal region was performed using CT vox software v3.3 (Bruker micro-CT). Color chromatic aberrations were adjusted to present a clear and analyzable image of the gut. Analysis was performed by a single experienced technologist to avoid reader variability in image analysis.

Histopathological evaluation and characterization of tumor lesions, and immunohistochemical examination. All colorectal tissues were dissected longitudinally and prepared as formalin-fixed, paraffin-embedded blocks. Serial sections of 5 μm were obtained using a rotary microtome (Accu-Cut SRM 200, Sakura Finetek, CA, USA). H&E staining was performed with DRS 2000 Automated Slide Stainer (Sakura Finetek) at room temperature after the routine protocol of deparaffinization, rehydration, and staining with hematoxylin solution for 5 min, followed by 5 dips in 1% HCl in 70% ethanol. Before mounting sections on glass slides, sections were rinsed, stained with eosin solution for 3 min, dehydrated with graded alcohol, and cleared in xylene. Tumor dimensions were measured using microcalipers, and tumor size was expressed as area calculated from the longest length and its maximum perpendicular width (length × width) [32].

Expressions of cytokeratin 20 (KRT20), CD3 protein (CD3), CD4 protein (CD4), and forkhead box protein 3 (FOXP3) were examined on an automated BenchMark GX slide stainer (Roche Diagnostics, Rotkreuz, Switzerland) with a closed and fixed program: deparaffinization at 75 °C for 8 min using EZ Prep solution (cat. no. 950-102, Ventana Medical Systems, Inc., Tucson, AZ, USA), antigen retrieval at 95 °C for 64 min (for KRT20, CD4, and FOXP3) or 92 min (for CD3) using Cell Conditioning 1 solution (cat. no. 950-124, Ventana Medical Systems), incubation at 37 °C with anti-KRT20 (1:400; cat. no. 18306-1-AP, Proteintech Group, Inc., Rosemont, IL, USA) for 1 h, anti-CD3 (clone 2GV6, prediluted; cat. no. 790-4341, Roche Diagnostics) for 2 h, anti-CD4 (1:200; cat. no. ab183685, Abcam, Cambridge, UK) for 2 h, or anti-FOXP3 (1:50; cat. no. ab 215206, Abcam) for 2 h. N-Histofine Simple Stain Mouse MAX PO (R) anti-rabbit (cat. no. 414341F, Nichirei Biosciences, Inc., Japan) was used as secondary antibody at 37 °C for 12 min twice, and visualization was performed by OptiView DAB IHC Detection Kit (cat. no. 760-700, Roche Diagnostics). All sections were counterstained with Hematoxylin II (cat. no. 790-2208, Ventana Medical Systems) at 25 °C for 8 min and with Bluing Reagent (cat. no. 760-2037, Ventana Medical Systems) at 25 °C for 4 min [29].

Finally, all sections were washed, mounted with an automatic cover slipper (Glas-J1; Sakura Finetek), observed, and photographed with an Echo Revolve microscope (RVL-100-M, Echo, San Diego, CA, USA). A medically qualified specialist examined the sections under high magnification to determine the cells with positive signals, which were quantified by using QuPath (version 0.3.0, https://qupath.github.io; accessed on 15 June 2023) [33].

Statistical analysis. The Mann–Whitney *U* test was used to compare tumor cover percentages and dimensions in CRC-bearing mice without and with propolis administration, and positive percentages of CD3^+^, CD4^+^, and FOXP3 lymphocytes in tumor microenvironments. These statistical analyses were computed using SPSS software (version 20, IBM Corp., Armonk, NY, USA), and the statistical significance was defined as *p* < 0.05.

## 3. Results

### 3.1. Propolis Administration Increased the Body Weight of DMH/DSS-Treated Mice

The body weight changes of the mice were used to evaluate their health status from the time of DMH administration [30]. The mice in both DMH/DSS-treated groups (Groups II and III) had significant weight loss after DMH/DSS induction. Two typical periods of weight loss occurred after the mice drank 3% DSS, but they resumed weight gain when they stopped drinking 3% DSS. In contrast, the mice gained weight only in the DMH/DSS-treated group with propolis administration (Group III) but not in the propolis-untreated group (Group II) (Figure 1B). These propolis- and DMH/DSS-treated mice in Group II always maintained a more stable body weight during the experimental time and showed a significantly increase at Day 20 (*p* = 0.039, Mann–Whitney *U* test).

### 3.2. Lower Contrast of µCT Imaging and Longer Length of Mice Colons in CRC Bearing Mice with Propolis Administration

To examine the DMH/DSS-induced changes in colons in vivo, µCT-images with different visual contrasts were acquired 3 h after the injection of an intravenous contrast agent (Figure 2). Briefly, the control mice with no DMH/DSS induction (Group I) showed a negative accumulation of barium in their colons. However, the colons of DMH-induced mice (Groups II and III) were easily visualized after the mice drank a second 3% DSS suspension, but a colon image with a relatively lower contrast was observed from the mice (Group III) further administered with propolis.

The colon length of the mice in the control group with no treatments (Group I) was 8.9 ± 0.1 cm (Figure 3). However, there was a decreased colon length in the DMH/DSS-treated mice (Group II) (7.9 ± 0.4 cm, Figure 3), while it was significantly recovered to 8.8 ± 0.2 cm in Group III (*p* = 0.046, Mann–Whitney *U* test). Moreover, in Figure 3, compared with the DMH/DSS-treated mice (Group II), the tumor area of the mice with DMH/DSS-induced CRC and administration of propolis (Group III) was relatively smaller.

### 3.3. Changes of Total Tumor Dimension in Colon of CRC-Bearing Mice with Propolis Administration

In our model of mice CRC induced by DMH/DSS for one month, the CRC tissues grew locally in the mucosa layer and had not grown beyond the muscularis mucosa of the colon (Figure 4). These early tumors were not found to have spread to nearby lymph nodes or to distant organs, no matter whether the mice were administered propolis or not. Moreover, the representative image in Figure 4A and the tumor dimensions in Figure 4B showed that the H&E-stained CRC tissue sections from the mice without propolis administration (Group II) had a larger tumor size (21.6 ± 7.4 mm^2^) than those in the propolis-administered mice (Group III) (7.1 ± 2.0 mm^2^) (Mann–Whitney *U* test, *p* < 0.05).

### 3.4. Effect of Propolis on the Level of KRT20 in CRC-Bearing Mice

The colon sections of the control mice without CRC (Group I) showed gradually increased signals of anti-KRT20 staining from the crypt bottom to the top of colorectal crypts (Figure 5). In contrast, the colon sections from the groups with DMH/DSS-induced CRC (Groups II and III) had intense staining from the anti-KRT20 antibody, but a moderate positive reaction was observed in the mucosal layer of the CRC tissues of the mice (Group III) with DMH/DSS induction and propolis administration (Figure 5).

### 3.5. Associations between the Propolis Administration and the Density of Tumor-Infiltrating T Lymphocytes Subsets in Mice with DMH/DSS-Induced CRC

To detect the differences of tumor-infiltrating T lymphocytes (TILs) in the gut microenvironment between the CRC-bearing mice without and with propolis administration, the signal intensities due to CD3, CD4, and FOXP3 in the lamina propria with neoplastic lesions were detected and measured (Figure 6). Briefly, the CRC-bearing mice (Group II) had a weakly positive signal and a lower density of CD3^+^ and CD4^+^ TILs in the lamina propria when mice were not administered propolis (Figure 6A). In contrast, the signal intensity and density of these TILs were significantly increased in the neoplastic lesions of CRC-bearing mice with propolis administration (Group III) (Figure 6B). Conversely, fewer FOXP3 lymphocytes were found in the lamina propria of the colons of mice in Group III when compared with the results from the CRC-bearing mice without propolis administration (Group II) (Figure 6A,B).

## 4. Discussion

We found that propolis has an anticancer effect on early stage CRC by affecting epithelial differentiation and gut immunity in the tumor microenvironment [34]. Recently, propolis has been found to have potential against SARS-CoV-2 infection and the resulting COVID-19 disease [35]. The anti-inflammatory properties attributed to propolis have also been examined in numerous studies [36]. For example, propolis is known to potentially treat or prevent gastrointestinal disorders, such as inflammatory bowel disease [37,38]. These outcomes are consistent with our results showing that propolis may have an anti-colorectal carcinogenesis property. As reviewed by Chiu et al., numerous studies have demonstrated that propolis is effective against various types of cancer, including CRC, due to the presence of various phytochemicals in propolis [19]. This propolis-induced anti-CRC property may be caused by enhanced apoptosis or an increased efficacy of co-administered anticancer drugs [16,39,40].

Like the previous model of CRC induction in mice by DMH/DSS administration [29], the one-month induction in this study limited the colon tumor lesion to the mucosa layer, which was diagnosed as early stage CRC. In this early stage CRC, we further revealed that the administration of propolis appeared to consistently attenuate the effect of DMH/DSS on CRC induction. We readily observed this anticancer effect of propolis on CRC regression by µCT, a technique that can diagnose CRC in vivo [41,42,43]. Here, we demonstrated that barium (sulfate) could cover the irregular mucosa on the colonic surface of CRC, showing slightly protruding plaques on images according to the severity of the CRC. Clinically, a combined barium/CT image better shows the destruction of the tumor mucosa than CT alone [41]. We further showed that µCT could be used to assess the neoplastic colonic lesions in vivo, even in early stage CRC. Our results are consistent with the report of Halligan et al. who said that computed tomographic colonography is a useful screening test in patients with symptoms suggestive of CRC [43]. However, an appropriate pathological diagnosis after in vivo X-ray µCT examination is also required. Thus, different pathological tests need to be performed to evaluate or demonstrate that propolis also reduces the size of colonic neoplastic lesions through the expression of common tumor markers, such as KRT20.

KRT20 is a marker for intestinal epithelium differentiation [44,45,46,47]. Imai et al. reported that KRT20 expression was closely associated with invasive histological features and had prognostic significance [46]. In other words, KRT20 is more commonly expressed in CRC tissues and nodal metastasis [48]. CRC patients with poor prognosis show high levels of KRT20 [46,47]. We found that KRT20 was absent in the crypt bottom but gradually increased towards the top of the colorectal crypts in the control group (Group I). This distribution in the normal colon is identical to that reported by others [44,49]. We further revealed that the increased KRT20 in DMH/DSS-induced CRC decreased because of the propolis administration. As Tunca et al. reported, the expression of KRT20 is associated with CRC recurrence and survival rates [50]. Thus, our data possibly indicated that propolis may improve CRC outcomes. Taken together, the decreased KRT20 activity due to propolis in early stage CRC tissues may reduce the chance of cancer worsening, metastasis, or recurrence.

For years, propolis has long been considered to have anti-inflammatory properties and has been used as an immunomodulatory agent [51,52]. This natural product is also known to promote gut health by modulating gut immunity [53]. In this study, we found that propolis administration increased the density of CD3^+^ and CD4^+^ TILs and decreased the density of FOXP3 lymphocytes in the lamina propria of the tumor microenvironment in the mice with early stage CRC. Intratumoral-infiltrating CD3, CD4, and FOXP3 lymphocytes are known to be strong positive predictive markers for the prognosis of CRC [54,55]. However, there are numerous reports showing that the FOXP3 lymphocytes in CRC are substantially different from those in other human cancers. For example, Kuwahara et al. reported that CRC patients with low CD4 lymphocytes and low FOXP3 lymphocytes exhibited extremely poor prognoses [55]. Ohue et al. further asserted that the presence of high numbers of FOXP3 lymphocytes corresponded to a better prognosis for CRC patients [56,57,58]. FOXP3 lymphocytes indicate a CRC prognosis, but in other cancers the presence of high FOXP3 lymphocytes in the tumor microenvironment is associated with unfavorable prognoses [59]. Thus, these variable differences must play a key role in the immunity of the gut microenvironment.

These contradictory results have stemmed from an assessment of the prognostic impact of FOXP3 lymphocytes in different tumor tissues [60,61]. These diagnostic difficulties are like those described by Betts et al., stating that the role of FOXP3 lymphocytes possibly depends on the type of immune response present in the tumor microenvironment, especially for CRC [60]. It is well known that CD3^+^ and CD4^+^ TILs in CRC correlate with a better prognosis, and that a high density of different TILs in the gut microenvironment is critical for immunity in the gut [54,55,62]. Therefore, propolis, which could increase the number of CD3^+^ and CD4^+^ TILs in the tumor microenvironment, can be classified as a natural product that is beneficial in preventing the CRC progression at the initial stage. Further, the FOXP3 lymphocytes could suppress the CD4^+^ TILs and so exacerbate CRC [60]. In other words, the decreased level of FOXP3 lymphocytes caused by propolis administration in the early stage of CRC formation could promote the increased level of CD4^+^ TILs to maintain gut immunity. As reported earlier by Terzic et al., CRC is enriched with abundant bacteria in the gut while malignancies in other tissues are in an almost sterile microenvironment [63]. These gut bacteria can infiltrate the tumor through the necrosis or ulceration of the colon surface to trigger inflammatory responses [64,65]. Xu et al. speculated that FOXP3 lymphocytes can suppress the inflammatory and immune responses caused by bacterial invasion, and thus may have a potentially anti-CRC effect [65]. However, this inference may be applied in advanced CRC stages with an erosive mucosa layer and invasive muscularis mucosa. In this study, the intact muscularis mucosa in mice was detected after inducing CRC with DMH/DSS for a month. This implies that the gut bacteria-induced inflammatory or immune response may not be obvious here, and that the immune suppression by FOXP3 lymphocytes at other advanced stage CRCs was not critical for the initial CRC formation.

Death rates for CRC have risen worldwide. Therefore, early detection and treatment are very important for preventive CRC medicine [66]. CRC is preventable at the very early precursor stage if the tumor cells are promptly suppressed as soon as they appear in the colon. Natural treatments that are harmless are now needed more than ever for the prevention of CRC [67]. As reported by Forma and Bryś, propolis and its components affect the tumor microenvironment and decrease the multidrug resistance of cancer cells [17]. Overall, and also given our current results, propolis can be considered as a nutritional supplement to improve human health [68].

In conclusion, propolis—a natural substance—could change the characteristics of CRC cells and improve the body’s immunity when CRC is in its early stages. Propolis potentially prevented CRC progression by increasing the levels of CD3^+^ and CD4^+^ TILs and reducing the levels of FOXP3 lymphocytes in the tumor microenvironment in early stage CRC. Moreover, our study indicates a promising role for propolis in complementary medicine, and it could be used as a food additive to prevent CRC progression. At the same time, studying the pathogenesis of carcinogen-induced CRC in mouse models has the advantage of being rapid and reproducible. However, it is difficult to compare different models with each other and directly apply our current results to the more complex human CRC [69,70]. These are limitations to this study that should be noted. First, the present study has not completely elucidated the precise mechanism by which propolis improves the gut immunity in the tumor microenvironment in CRC. Second, we only used a single brand of propolis, and the biological activity of propolis depends on its particular extraction techniques [71]. Third, the anticancer efficacy, adverse events, and toxicity of propolis must be evaluated through more immune confirmation and human trials. Taken together, a better understanding of the relationship between propolis and its anti-CRC effects should be clarified by further investigations.

## Figures and Tables

**Figure 1 nutrients-15-04494-f001:**
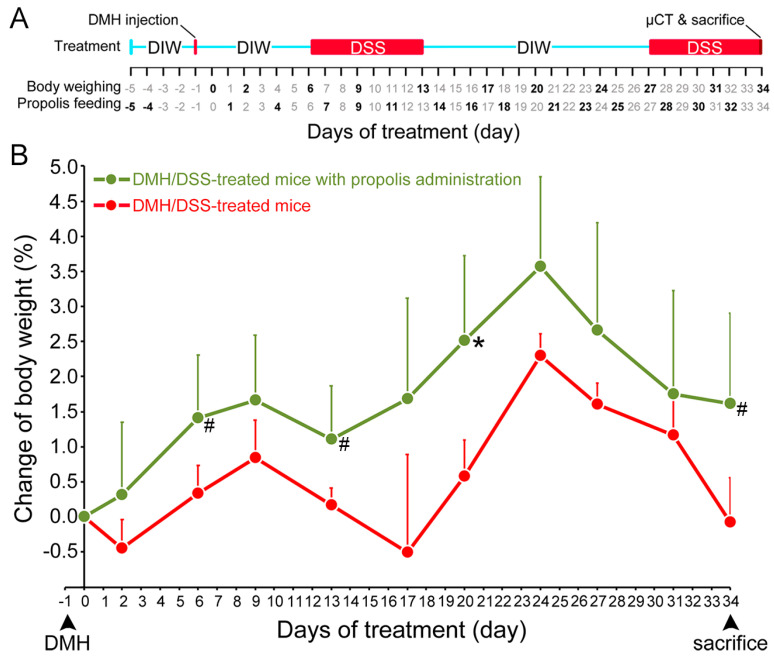
Change in body weight of CRC-bearing mice following propolis administration. (**A**) Timing of DMH and DSS induction of CRC and propolis administration. Red line indicated the day for subcutaneous injection of DMH (40 mg/kg body weight) and red boxes were the days for drinking DSS (3%). The days for body weighing and propolis (30 mg/mouse) feeding were indicated in bold. The X-ray microcomputed tomograph (µCT), sacrifice, and colon sampling were performed at day 34. (**B**) Body weights. Red line, mice with DMH/DSS treatment; green line, mice with DMH/DSS treatment and propolis administration. CRC, colorectal cancer; DMH, 1,2-dimethylhydrazine; DSS, dextran sulfate sodium; DIW, deionized water. *, *p* < 0.05 and #, 0.05 < *p* < 0.1 (Mann–Whitney *U* test).

**Figure 2 nutrients-15-04494-f002:**
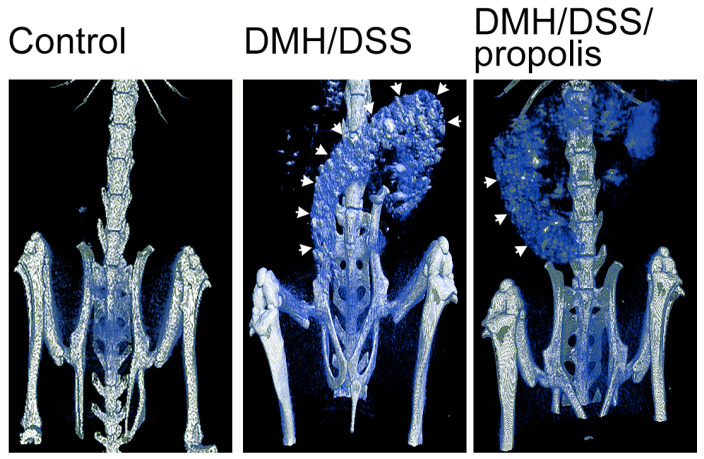
Detection of CRC in mice by X-ray microcomputed tomograph. Each mouse was imaged 3 h after administration of Baritop LV (30 mg/kg). Control mice, not treated with DMH/DSS and not given propolis; DMH/DSS, mice with DMH (40 mg/kg body weight)/DSS (3%) treatment; DMH/DSS/propolis, mice with DMH/DSS treatment and propolis administration. White arrows indicated the sites with barium accumulation. CRC, colorectal cancer; DMH, 1,2-dimethylhydrazine; DSS, dextran sulfate sodium.

**Figure 3 nutrients-15-04494-f003:**
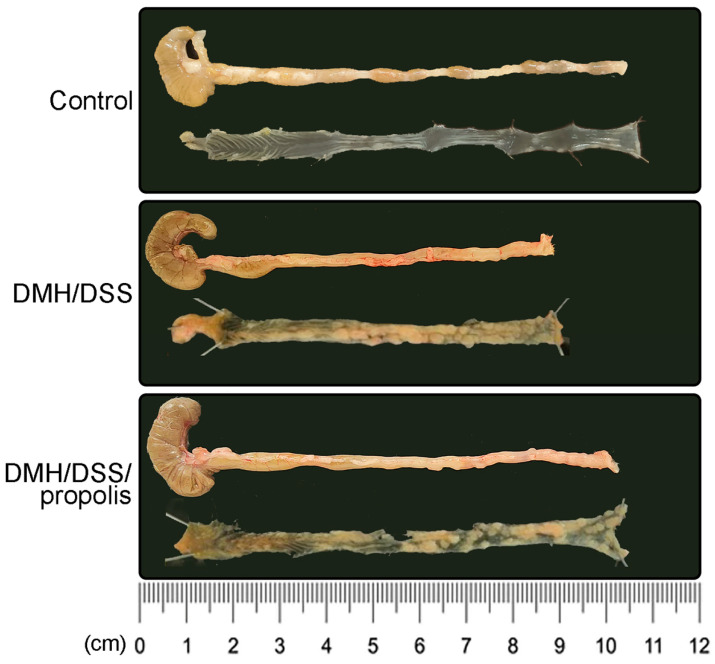
Inhibition of colon shortening in CRC-bearing mice by administration of propolis. Each group was individually framed with a black box including the representative colon image (upper one) and its inner layer (lower one). CRC, colorectal cancer; DMH, 1,2-dimethylhydrazine; DSS, dextran sulfate sodium.

**Figure 4 nutrients-15-04494-f004:**
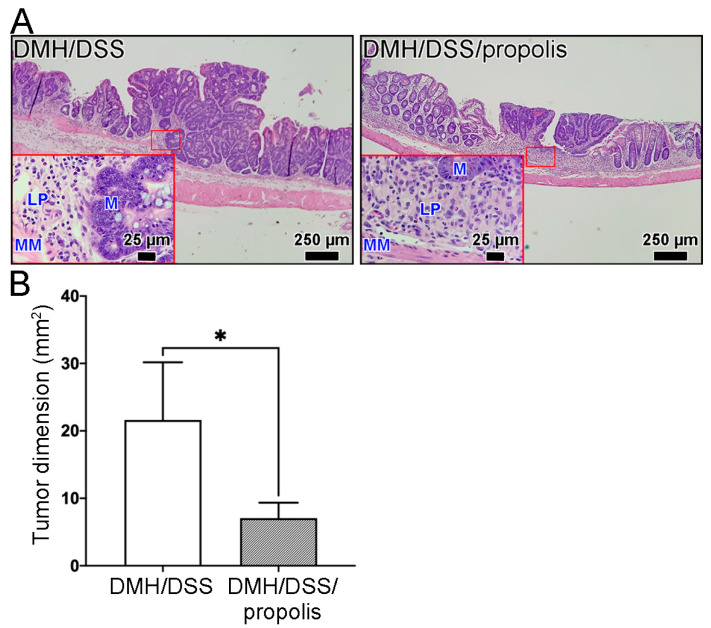
Reduction in DMH/DSS-induced CRC formation following propolis administration. (**A**) Representative histopathological images of mice colons. Scale bar, 250 µm. Red square indicated an image with a higher magnification (scale bar, 25 µm). (**B**) The tumor dimensions from the H&E-stained CRC tissue sections. CRC, colorectal cancer; DMH, 1,2-dimethylhydrazine; DSS, dextran sulfate sodium; M, mucosa; LP, lamina propria; MM, muscularis mucosa; H&E, hematoxylin and eosin. *, *p* < 0.05 (Mann–Whitney *U* test).

**Figure 5 nutrients-15-04494-f005:**
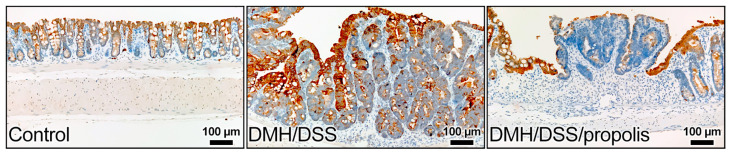
Representative images of immunohistochemical staining for KRT20 in colon of mice. KRT20 staining increased from bottom to the top of the crypt in colonic mucosa of control mice. An intense staining of KRT20 was in the colon sections from the mice with DMH/DSS-induced CRC and a moderate positive reaction was in the mucosal layer of CRC tissues of the mice with propolis administration. KRT20, cytokeratin 20; CRC, colorectal cancer; DMH, 1,2-dimethylhydrazine; DSS, dextran sulfate sodium. Scale bar, 100 µm.

**Figure 6 nutrients-15-04494-f006:**
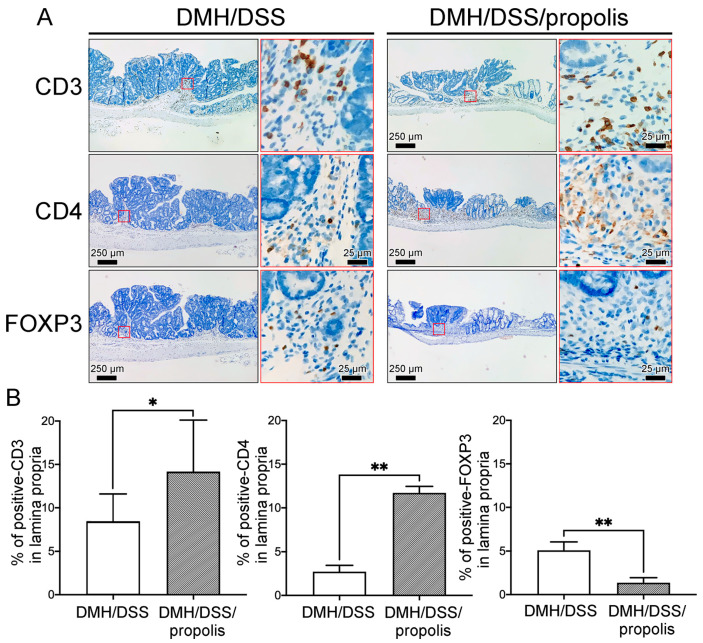
Effects of propolis administration on the density of tumor-infiltrating T lymphocytes in colons of CRC-bearing mice. (**A**) CD3^+^, CD4^+^, and FOXP3 lymphocytes in the lamina propria with neoplastic lesions in colons of CRC-bearing mice. Scale bar, 250 µm. Red square indicated an image with a higher magnification (scale bar, 25 µm). (**B**) The signal intensity and density of CD3^+^, CD4^+^, and FOXP3 lymphocytes. CRC, colorectal cancer; DMH, 1,2-dimethylhydrazine; DSS, dextran sulfate sodium. CD3, CD3 protein; CD4, CD4 protein; FOXP3, forkhead box protein 3. *, *p* < 0.05 and **, *p* < 0.01 (Mann–Whitney *U* test).

## Data Availability

The datasets used and/or analyzed during the current study are available from the corresponding author on reasonable request.

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
