# Peer review of "Propolis Has an Anticancer Effect on Early Stage Colorectal Cancer by Affecting Epithelial Differentiation and Gut Immunity in the Tumor Microenvironment"

_nutrients, 2023, doi:10.3390/nu15214494_

Round 1

Reviewer 1 Report

Comments:

The manuscript “Propolis has an anticancer effect on early-stage colorectal cancer by affecting epithelial differentiation and gut immunity in the tumor microenvironment” presented is comprehensive and provides a good amount of information on the topic. Here are some critical comments and suggestions for improvement:

·       In the abstract, the sentence "Our study could indicate a promising role..." is speculative. "Indicates" can be changed to "suggests" to make the statement less assertive.

·       The authors mention the westernization of diets as a contributing factor to colorectal cancer but don't provide evidence or delve deeper. The association between diet and colorectal cancer should be substantiated.

·       Introduce propolis's relevance and potential in colorectal cancer intervention in the introductory part.

·       The manuscript lacks information on the rationale for the chosen propolis dosage (30 mg/mouse per time) and the frequency of administration. The justification for this dosing regimen should be provided.

·       The duration of the experiment (about five weeks) seems relatively short for assessing the long-term effects of propolis on CRC progression. Discuss the rationale for this duration and its relevance to the clinical scenario.

·       The sample sizes in the study groups are very small (2-3 mice per group). Such a small sample size reduces the study's statistical power and increases the likelihood of type II error. It would be better to have a larger sample size for robust conclusions.

·       It's unclear why only BALB/c mice were chosen; a rationale should be provided.

·       Provide information on validating the imaging techniques (μCT) used for assessing colonic changes. Mention any potential limitations or sources of bias in the imaging and analysis process.

·       Consider discussing the limitations and potential sources of bias in the study, as well as the implications of the results for future research or clinical applications.

·       Considering this is a mouse model study, caution should be exercised when generalizing the findings to human CRC. The discussion or conclusion could reflect this.

·       Discussing any potential side effects or interactions of propolis might be essential, even if it's generally considered safe. Also, the conclusion mentions propolis as a "harmless substance," but it is essential to highlight potential safety concerns or side effects associated with propolis supplementation, as well as the need for further safety studies.

·       The discussion is filled with textbook explanations rather than a proper discussion, which can be improved.

·       Lastly, thorough proofreading would be beneficial, as there are a few grammatical errors.

A thorough proofreading and language editing would be beneficial, as there are a few grammatical.

Author Response

  1. In the abstract, the sentence "Our study could indicate a promising role..." is speculative. "Indicates" can be changed to "suggests" to make the statement less assertive.

Response: Thanks for your professional comment. We have replaced “indicate” with “suggest” in the last sentence of Abstract section.

  1. The authors mention the westernization of diets as a contributing factor to colorectal cancer but don't provide evidence or delve deeper. The association between diet and colorectal cancer should be substantiated.

Response: We cited 4 references (the original references #4 to #7) to indicate that “incidence of young people suffering from CRC is rising globally due to the westernization of diets”. However, the following sentence has been added to the first paragraph of Introduction section: The western diet may have deleterious effects on the gut, such as promoting the unfavorable microbes, increasing DNA damage in colon cells, and impeding their DNA repair (the original references #4 and #7) (lines 44-45, in the newly edited version).

  1. Introduce propolis's relevance and potential in colorectal cancer intervention in the introductory part.

Response: We deeply appreciate the reviewer’s comment. Although we have discussed the potential correlation between propolis and CRC in the 2nd paragraph of Discussion section, we now further add the following sentences to state the critical importance of propolis in CRC at the end of the 2nd paragraph of Introduction section: This natural product not only exhibits potent anti-CRC activity by regulating various signaling molecules (reference #19), but also triggers CRC cell death by increasing DNA condensation to reduce the proliferation rate (references #20 and #21) (lines59-61, the newly edited version).

  1. The manuscript lacks information on the rationale for the chosen propolis dosage (30 mg/mouse per time) and the frequency of administration. The justification for this dosing regimen should be provided.

Response: We must apologize our carelessness. (i) The dosage of propolis used in this study was based on the report (reference #31) of Castro and Higashi who demonstrated 1.2 g propolis/kg per day by oral administration. In this way, each 25-gram mouse is fed approximately 30 mg of propolis. (ii) We hoped to reduce the pain and discomfort caused by gavage in mice by reducing the number of feedings, thereby achieving the purpose of 3Rs (Replacement, Reduction and Refinement). In other words, we chose to feed once every two days instead of every day.

  1. The duration of the experiment (about five weeks) seems relatively short for assessing the long-term effects of propolis on CRC progression. Discuss the rationale for this duration and its relevance to the clinical scenario.

Response: This study was originally designed to explore whether propolis, a natural product, can change the possibility of early-onset CRC. Therefore, based on our previous animal model of DMHDSS-induced CRC, mice would develop early CRC within the first 4–6 weeks of DMH/DSS induction (Oncol Lett 2020 20(6):327; Int J Oncol 2022 60(6):64). This is why we only analyzed the relative short term here.

  1. The sample sizes in the study groups are very small (2-3 mice per group). Such a small sample size reduces the study's statistical power and increases the likelihood of type II error. It would be better to have a larger sample size for robust conclusions.

Response: Thanks for reviewer’s concern and comment. We must apologize that sample size of mice with DMH/DSS treatment (Group II) is 6, not the original 3. However, it is still a small sample size. We know that the large sample size is good to generate reliable data. However, we have to achieve the goal of 3Rs (Replacement, Reduction and Refinement), and our data is consistent even if the number of each group is not large.

  1. It's unclear why only BALB/c mice were chosen; a rationale should be provided.

Response: From our previous studies (the original references #29 and #30), the Balb/c mice are the better model to induce CRC with DMH/DSS.

  1. Provide information on validating the imaging techniques (μCT) used for assessing colonic changes. Mention any potential limitations or sources of bias in the imaging and analysis process.

Response: Barium is a pure nonradioactive substance that can be used as a contrast agent in radiology and fluoroscopy. Clinically, a combined barium/CT image will be better to show the destruction of tumor mucosa than CT alone. It is why we used barium and µCT to analyze the potential tumor lesion in colon in vivo. As we stated in the 2nd paragraph of Discussion section, the “barium (sulfate) could cover the irregular mucosa on the colonic surface of CRC, showing slightly protruding plaques on images according to the severity of the CRC. However, as reported by Halligan et al., the computed tomographic colonography may be a screening test in patients with symptoms suggestive of CRC. Therefore, an appropriate CRC pathological diagnosis is needed after in vivo X-ray µCT detection of mice. Furthermore, we add the statement and two new reference (references #40 and #42) in the 2nd paragraph of Discussion section of the revised version (lines 317 to 324, in the revised version).

  1. Consider discussing the limitations and potential sources of bias in the study, as well as the implications of the results for future research or clinical applications.

Response: We discuss the limitation as followings and have added to the final of Discussion section. “…There are limitations to this study that should be noted. First, the present study may not have completely elucidated the precise mechanism by which propolis improves the gut immunity in the tumor microenvironment in CRC. Second, we only used a single brand of propolis. However, the biological activity of propolis depends on different extraction techniques. Third, the anticancer efficacy, adverse events, and toxicity of propolis must be evaluated through more immune confirmation and even human trials. Taken together, a better understanding of the relationship between propolis and its anti-CRC effects should be clarified by further investigations”. Furthermore, we add one new reference (reference #70) in the revised version.

  1. Considering this is a mouse model study, caution should be exercised when generalizing the findings to human CRC. The discussion or conclusion could reflect this.

Response: Studying the pathogenesis of carcinogen-induced CRC in mouse models has the advantage of being rapid and reproducible. However, it is difficult to compare different models with each other and directly apply our current results to the more complex human CRC. This point and two new references (references #68 and #69, respectively) have been added in the new Discussion section.

  1. Discussing any potential side effects or interactions of propolis might be essential, even if it's generally considered safe. Also, the conclusion mentions propolis as a "harmless substance," but it is essential to highlight potential safety concerns or side effects associated with propolis supplementation, as well as the need for further safety studies.

Response: Thanks for the reviewer’s reminding that harmless may be arbitrary. (i) We firstly deleted the word “harmless” in the penultimate sentence of the Abstract section. (ii) We also have deleted the word “harmless” and replaced it with the word “natural” in the first sentence of the last paragraph of Discussion section. (iii) As reviewed by Miguel et al., propolis is a complex natural product but not entirely innocuous. The source and quality control of propolis during production are crucial to its efficacy and human health. Thus, further safety studies is necessary and relevant statements are included at the end of new Discussion section (lines 394-400, in the revised version).

  1. The discussion is filled with textbook explanations rather than a proper discussion, which can be improved.

Response: Thanks for reviewer carefulness. Now, we have revised parts of Discussion section at the first three paragraphs to focus on discussing our results. Finally, the last paragraph contains limitations to clearly point out the need for future works.

Reviewer 2 Report

In the manuscript, Shen et al. attempted to investigate the anticancer effect of propolis on early-stage colorectal cancer. The main concern for me is the statistical analysis. There are three groups in the experiment, and ANOVA should be more suitable to compare the differences among them, instead of student t test. In ALL the comparisons, it seems that the authors totally got rid of the Group I, as shown in Figure 1, 4 and 6. The authors should explain this.

Moreover, several minor revisions should be considered, as follows,

1.     Line 28 were changed or were significantly changed?

2.     Line 43 Can people whose age is below 50 be considered as young people?

3.     In Materials and Methods section, the subtitles should be used to make the structure clear.

4.     Line 90 and Line 208 CO2 and mm2 should be “CO2” and “mm2”.

5.     Line 175-179 What is the statistically comparison between Group II and III?

6.     Line 156 The statistically significances should be indicated in Figure 1.

7.     Line 203 Please substitute “mouse” by “mice”.

8.     The discussion section should be re-ranged to make the logical line clear.

Minor editing of English language required.

Author Response

In the manuscript, Shen et al. attempted to investigate the anticancer effect of propolis on early-stage colorectal cancer. The main concern for me is the statistical analysis. There are three groups in the experiment, and ANOVA should be more suitable to compare the differences among them, instead of student t test. In ALL the comparisons, it seems that the authors totally got rid of the Group I, as shown in Figure 1, 4 and 6. The authors should explain this.

Response: Thanks to the reviewer for the reminding. As suggested by other reviewers, we have changed our statistic method to the Mann–Whitney U test, a nonparametric method. This statistical method may be more suitable for our current study. The symbols of P-values in Figures 1, 4, and 6 and their legends have been also corrected in the revised version.

Although we initially designed three groups (group of normal control without any treatment, group of DMH/DSS induction, and group of DMH/DSS induction and further propolis administration), data of control group, as image results, were only shown in Figures 2, 3, and 5. Followings are the reasons why Figures 1, 4, and 6 did not show the control group data. (i) In fact, we focused on the change of body weight of mice with CRC after taking propolis in Figure 1. This analysis therefore targeted mice with DMH/DSS-induced CRC. (ii) In addition, we counted and calculated the tumor dimension (Figure 4) and percentages of T-cells with positive for immune markers in tumors (Figure 6). No tumors were detected in the control mice, so no tumor imaging data could be available and presented here.

Moreover, several minor revisions should be considered, as follows,

  1. Line 28 were changed or were significantly changed?

Response: Thanks for reviewer’s comment and we have changed it.

  1. Line 43 Can people whose age is below 50 be considered as young people?

Response: Thanks for reviewer’s carefulness. Now we have deleted the word “young” and changed this sentence to: …the incidence of people under 50 years of age suffering from CRC…

  1. In Materials and Methods section, the subtitles should be used to make the structure clear.

Response: We have retitled the subtitles in the section of Materials and Methods as followings: “CRC carcinogenesis, animals, and CRC induction” to “Experimental animals and CRC induction by carcinogen”; “In vivo µCT, µCT 3D reconstruction, and image analysis” to “In vivo µCT" and image analysis”; “H&E staining for histopathological evaluation and characterization of tumor lesions, and immunohistochemical examination” to “Histopathological evaluation and characterization of tumor lesions, and immunohistochemical examination”.

  1. Line 90 and Line 208 CO2 and mm2 should be “CO2” and “mm2”.

Response: Thanks for reviewer’s carefulness and we have corrected them.

  1. Line 175-179 What is the statistically comparison between Group II and III?

Response: The P-value of comparison between Groups II and III is 0.046 by the Mann–Whitney U test, a nonparametric method. Further, we have added it to and revised the corresponding sentence in the Results section of the revised version (lines 193-196).

  1. 6.Line 156 The statistically significances should be indicated in Figure 1.

Response: We have added the statistically significance in the revised Figure 1 and also stated this in the first paragraph of Results section (lines 168-169, in the revised version).

  1. Line 203 Please substitute “mouse” by “mice”.

Response: Thanks for reviewer’s comment and we have replaced it.

  1. The discussion section should be re-ranged to make the logical line clear.

Response: Integrate other reviewers’ suggestions, the Discussion section has been improved. Now, we have revised parts of Discussion section at the first three paragraphs to focus on discussing our results. Finally, the last paragraph contains limitations to clearly point out the need for future works.

Reviewer 3 Report

The paper "Propolis has an anticancer effect on early-stage colorectal can-2 cer by affecting epithelial differentiation and gut immunity in 3 the tumor microenvironment" is a research that deserves publication only due to the commitment and professionalism demonstrated by the authors, however, in my opinion there are some observations to make:

A) the authors talk about CRC but in reality from the images produced and from what has been written the induced lesions are limited to the mucosa and therefore, being above the muscularis mucosae, they have no metastatic power and therefore it would be more correct to define them as low and high grade dysplastic lesions.

B) the activity of propolis in blocking neoplastic growth was evaluated with antibodies against CK 20, Cd3 Cd4 and Foxp3 which I think are a bit limiting for example the antibody against CK7 and CDX2 could also have been used and in any case judging the blocking power of propolis only by the expression of these molecules is, I repeat, limiting and needs to be better described.

Minor editing.

Author Response

A) the authors talk about CRC but in reality from the images produced and from what has been written the induced lesions are limited to the mucosa and therefore, being above the muscularis mucosae, they have no metastatic power and therefore it would be more correct to define them as low and high grade dysplastic lesions.

Response: As shown in our previous published reports (J Adv Res. 2019 22:7-20 and Oncol Lett. 2020 20(6):327), using the model of DMN/DSS-induced CRC in mice, CRC will be in the early stage during the first month of induction. Thus, the cancer cells are only in the mucosa layer. This is also in line with the purpose of this study to explore the use of propolis for prevention of early CRC.

B) the activity of propolis in blocking neoplastic growth was evaluated with antibodies against CK 20, Cd3 Cd4 and Foxp3 which I think are a bit limiting for example the antibody against CK7 and CDX2 could also have been used and in any case judging the blocking power of propolis only by the expression of these molecules is, I repeat, limiting and needs to be better described.

Response: Thanks for reviewer’s comments. (i) We know that the markers for immune detection here are not comprehensive, and there is indeed for more additions, such as CK7 and CDX2 which are suggested by the reviewer. However, due to limitations at this stage, we cannot fully carry out it. We will gradually complete and add these in the future and this limitation has beeni stated in the last paragraph of Discussion section of the revised version. (ii) As mentioned in the previous reply, other limitations have been added to the discussion section.

Reviewer 4 Report

The presented knowledge is novel, the conducted study demonstrates high scientific quality and is both necessary and relevant to the advancement of nutritional science.  My evaluation is that the manuscript is well-prepared and, with the following suggestions taken into account, suitable for publication in the Nutrients Journal.

1)      The introductory section explains the study design. The Authors justify the research topic well.

2)      Methodology section - I request clarification from the Authors on whether they checked the normality of the obtained results' distribution. Typically, comparable results assume a deviation from normal distribution, requiring data transformation for the parametric test of the Student's-T test.

3)      The description of the results is correct and in line with the usual description of manuscripts.

4)      The conclusions are well formulated and fully relate to the results obtained.

5)      The formatting of the text and the layout of the manuscript are correct, according to the requirements of the journal.

Author Response

  • The introductory section explains the study design. The Authors justify the research topic well.

Response: Thanks for reviewer’s affirmation.

  • Methodology section - I request clarification from the Authors on whether they checked the normality of the obtained results' distribution. Typically, comparable results assume a deviation from normal distribution, requiring data transformation for the parametric test of the Student's-T test.

Response: We appreciate reviewer’s comment. It is inappropriate to apply Student’s t test here. In the revised version, we have used the Mann–Whitney U test, a nonparametric method, to replace the original Student’s t test and yield the significant P-values. Further, the symbols of P-values in Figures 1, 4, and 6 and their legends have been also corrected in the revised version.

  • The description of the results is correct and in line with the usual description of manuscripts.

Response: Thanks for reviewer’s affirmation.

  • The conclusions are well formulated and fully relate to the results obtained.

Response: Thanks for reviewer’s affirmation.

  • The formatting of the text and the layout of the manuscript are correct, according to the requirements of the journal.

Response: Thanks for reviewer’s affirmation.

Round 2

Reviewer 3 Report

The authors responded adequately to the suggestions proposed and therefore the paper is worthy of being published in its current version.